# CSR and the Hermeneutical Renovation of Foucault's Toolbox

## Jeremy Tauzer

Philosophy Department, Saint Louis University, St. Louis, MO 63108, USA; jeremiah.tauzer@slu.edu

**Abstract:** This article aims to examine Foucault's conceptual toolbox (methodology, conceptual tools, and conceptual meta-tools) in relation to the socio-historical analysis of CSR and of the corporation. The article has a bidirectional purpose: it aims to use Foucault's toolbox to analyze CSR, and to use the occasion of applying Foucault to CSR to reflect on the interpretation, critical potential, and adequacy of Foucault's conceptual toolbox. It starts with some preliminary work: a review and rework of an interpretation of Foucault's conceptual toolbox by Koopman and Matza. With this interpretationally revised toolbox in mind, it then initiates a Foucauldian approach to the research field of 'the corporation' and the sub-field of CSR. Most of the first half of this article demonstrates that Foucault's toolbox offers a fruitful start to tackling these fields. The second half of the article takes up a counterpoint in the reverse direction, namely that Foucault's toolbox is not equipped for adequately apprehending the interpretative play and flexibility operating within CSR discourse. This leads to a suggestion of three ways to incorporate hermeneutic tools into Foucault's toolbox, and to an exemplification of how such toolbox renovation sheds new light on the tactics and power dynamics of CSR discourse.

**Keywords:** Foucault; CSR; corporation; dispositive; hermeneutics; concepts; methodology; discourse

## 1. Introduction

The intention of this article is to explore, problematize, reconceptualize, and reinterpret what it looks like to engage Michel Foucault with the empirical research field of CSR.

The methodology of this article can be described as hermeneutic. This article intends to transact primarily in concepts and methodologies rather than particulars—it operates mainly at the level of the play of theories and concepts and their potential to challenge and reinterpret each other. The empirical fields referenced are vital to this hermeneutic process, but this article abstains from engaging directly in empirical work, instead situating itself in a broadly parasitic manner on the back of the detailed empirical work of others. This abstention from empirical work leaves this article free to engage in a more abstract problematization, namely, the problematization of the encounter between Foucault's toolbox and CSR. This involves a bidirectional challenging: just as Foucault can be used to challenge and reinterpret the field of CSR, so also the contours of the CSR field can be used to challenge and reinterpret Foucault's 'conceptual toolbox' (a rephrasing of Foucault's term 'toolkit' [1] (p. 145)).

This methodology thus leaves open, and in fact welcomes, the possibility that more proper empirical work can challenge, improve upon, or add substance to its broad and conceptual claims.

In Section 2, I lay the groundwork for this bidirectional challenging between Foucault and CSR by attempting to give an accurate interpretation of Foucault's toolbox. Foucault's toolbox can roughly be conceived as the collection of methodologies and concepts which Foucault utilizes across his work. I do this by reviewing and reworking a helpful depiction of Foucault's toolbox by Colin Koopman and Tomas Matza [2]. This revised toolbox helps provide the context for then outlining, in broad strokes, a Foucauldian approach to the field of the corporation and the (sub)field of CSR (in Sections 3–5)—for utilizing Foucault

to approach and challenge various approaches to CSR and its broader context (the corporation). The basic strategy of the latter part of the article (Sections 6–8) is to take up the opposite hermeneutic direction: how do the contours of the CSR field challenge Foucault's toolbox and perhaps illustrate a need to renovate it? My claim is that the discourse about and around CSR have an interpretative flexibility—a hermeneutic spaciousness—which show an insufficiency in applying Foucault's traditional methods for analyzing discourse (Section 6). This leads me to suggest three ways to hermeneutically renovate Foucault's toolbox (Section 7), and to give a limited demonstration of the immediate usefulness of applying such renovations to the field of CSR (Section 8). My hope is that this gives the reader a starting point for exploring my claim that a hermeneutically renovated Foucault sheds new light on the tactics and power dynamics of CSR and its discourse.

## 2. Foucault's Toolbox

Recently, Hans Sluga has said of Ludwig Wittgenstein: "What makes him important and what will keep him in our view is not our agreement with his conclusions, but the fact that we can learn from the intensity and engagement of his thinking, from his techniques of questioning, from his never satisfied impulse to rethink the issues" [3] (p. 119).

What, then, about Foucault, "keeps him in our view"? One mustn't see Foucault as a system, or a theory, or a set of theories. Foucault voiced disdain for totalizing 'theory' [4] (p. 231), but rather, envisioned theory as a localized practice (p. 208). Additionally, he resisted the idea of maintaining a stable identity [5] (pp. xviii–xix). In short, he is not a regime. Rather, Foucault is a moving stream, a conceptual toolbox, a disruption, an activist, or a catalyst; possibly even an exercise, or a practice. Foucault had apprehensive self-consciousness about stepping into a stream of discourse in his inaugural lecture "The Order of Discourse" [6], and thus, rather than staying apt to a system or even to a method or set of tools offered by Foucault, it seems more apt for us to be conscious of our own discourse and conceptual tool development. Though Foucault's tools are handy and diverse enough to spend a lifetime studying, perhaps we would do better to be on the lookout for new tools as well as new applications for those tools. This requires a sort of openness or flexibility. However, what sort or sorts of flexibility should one entertain if one wants to utilize Foucault in new ways upon new fields of analysis?

A great place to turn to with such a question is a paper by Koopman and Matza titled "Putting Foucault to Work" [2]. In it, they do a helpful job at organizing the Foucauldian conceptual landscape. I will first take a little liberty in giving an interpretative (oversimplified) picture of their schematization (my depiction in Figure 1). Here's the picture: Foucault comes equipped with very broad tools to the object of his research. These tools are broadly useful across domains: these include "analytics" (or methods) such as archeology and genealogy, and "categories" such as power, knowledge, and the subject/self. The object of research, to which these tools will be applied, is basically a chosen "topic", which roughly provides the domain or field of research (such as psychiatry, punishment, or sexuality). This topic will need further organization into "sites, fields, and objects", such as the compilation of an archive. What emerges from this fieldwork in which meta-tools ("analytics" and "categories") encounter their objects ("topics", "sites", "fields", and "objects") is first concepts (such as discipline and governmentality), and then conclusions about historical developments and conditions. Another kind of element which emerges is a "doctrine": doctrines "refer to the philosophical results of Foucault's inquiries" (p. 822), and examples include "nominalism" and "historicism" (p. 824).

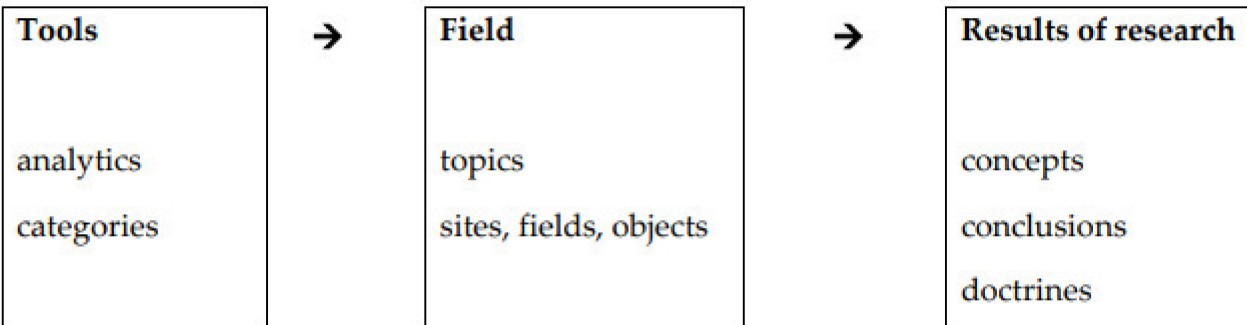

**Figure 1.** My depiction of Koopman and Matza's Foucauldian framework.

As incredibly useful as this schema is, especially to researchers in other fields who are trying to put some order into the Foucauldian chaos, my basic impression is that Koopman and Matza did not take to heart the sort of lesson that Sluga got out of Wittgenstein. Foucault had a dynamism which was not intended to be captured in a stable theory or a set system of analysis. It is that dynamism which made Foucault more open toward changing his own views and reinterpreting his earlier research in the light of his new perspectives, and which can also equip us to be more open to reinterpreting or renovating Foucault's whole toolbox (which I will do especially in Section 7). More specifically, I have three objections to Koopman and Matza's approach.

First, the idea that Foucault has "doctrines" which are the "philosophical results of Foucault's inquiries" is a painfully philosophical interpretation of someone who decidedly trying to stay away from systems and theories and doctrines—who was trying hard to "think otherwise" than in the old discursive grooves of disciplines like philosophy. Foucault's broadest comments about his work were presented more along the lines of practices or resistances—nominalism as a way of doing research which resists the power of universals, positivism as a way of thinking otherwise than ideological or hermeneutic or structuralism, a 'history of the present' as a way of resisting subjectivizations (ways in which subjects have been shaped) by unearthing the contingent conditions of their formation. In other words, I would rather think of nominalism, historicism, and empiricism—and even more vague pictures such as "a history of the present", critique, "thinking otherwise", and research as exploratory or experimental—as meta tools. Such meta-tools can be viewed as even broader than the referenced 'tools' of analytics and categories and can even be conceived as guiding in the sculpting of such tools. One can also ask whether Foucault's nominalism should be taken merely methodologically (how to research) or also philosophically (as doctrinal), but apparently Foucault did not prioritize giving clear answers to such questions. He was more interested in what his 'nominalism' and research could *do*: "What I say ought to be taken as . . . 'game openings' . . . they are not meant as dogmatic assertions . . . . My books aren't treatises in philosophy or studies of history: at most, they are philosophical fragments put to work in a historical field of problems" [7] (p. 74).

Secondly, Koopman and Matza create a directionality from a stable set of tools to a variable field and a variable collection of results which does not do proper justice to the empirical openness and experimental flexibility of Foucault's approach. (They do acknowledge that although Foucault's analytics are more stable and mobile than his "topics" and "concepts," they probably should not be taken as universal (p. 838). However, I am claiming that this is not enough with respect to capturing the experimental nature of Foucault's practice of research.) There was a multifaceted flexibility to Foucault's approach. First, although he would start with a 'topic' which struck him as problematic—such as the clinic, or the prison, or biopolitics, or the "Hermeneutics of the Subject"—he had a flexibility about the field itself. This was particularly manifest in how "The Birth of Biopolitics" and "The Hermeneutics of the Subject" hardly get around to their initial targeted problematic objects, because Foucault ended up extensively attending to problems (respectively, neoliberalism and care of the self) which formed the backgrounds for the

problematizations he originally meant to target. A second kind of flexibility was flexibility about the tools one applies to a field. For instance, it is hard to get a clear grasp of the precise nature of the shifting focus from 'archeology' to 'genealogy' in the 1970s, but it would be hard to deny that there was an important development in Foucault's tools of 'analytics'. Foucault's sense of categories also developed and adjusted alongside his migrations to new empirical fields. So, for instance, his engagement with the field of criminality roughly coincided with his development of the *power* category of analysis; his engagement with the field of the state roughly coincided with his development of *governmentality*; and his engagement with the field of ancient ethics roughly coincided with his development of the *subject* category. At one point, he even criticized his crucial conception of power knowledge: "the second shift in relation to this notion of power-knowledge involves getting rid of this in order to develop the notion of government by the truth" [8] (p. 47). New tools can open up new fields, but new fields can call for re-tooling and renovating one's tools. Foucault was not an objectivity-conscious social scientist (no one is perfect)—he can be accused of all the 'bias' opened up by admitting a bidirectional porosity between methodology and field.

My third and last objection is that the way Koopman and Matza present what I call "results of encounter" is highly tamed and academic. It is missing the fire which comes out far more often in Foucault's interviews than in his lectures [1,4]. This is the fire of an activist, who is looking for his research to be put 'to work' in the present not merely intellectually, but as weapons of resistance and struggle and escape (see, e.g., [9] (p. 493) for some quotes and discussion)—weapons for change. With Foucault's ethical turn, this eye to change the present remained, even as it took a more subject-oriented flavor—e.g., "it's up to us to advance a homosexual ascesis that would make us work on ourselves and invent—I do not say discover—a manner of being" [5] (p. 137).

In light of these three criticisms, I have illustrated my own renovation of Koopman and Matza's schema in Figure 2. The main changes are to remove 'doctrines' and instead add 'meta-tools' on the far left as a new part of the methodological toolbox, to replace the unidirectional arrows with bidirectional arrows, and to add 'tools for change' to the 'results' box. There are some additional changes to terminology, such as replacing 'tools' with 'methodological toolbox' so as to gesture at the idea that results of research can be tools of a different sort, replacing 'topic' with 'problematization', and replacing 'conclusions' with 'reconceived historical landscape'. I have added some spaces to signify that some of these elements are better conceived as having "diagonal" relationships rather than simply as equals on a list. Additionally, the spaces are meant to conform to a general pattern in which the more to the left, the more stable and general an element can be applied (meta-tools are most stable), whereas the rightmost are the most fluid and specific (though concepts do not really fit this pattern since they sometimes apply across different fields). Altogether, this schema sacrifices some of the simplicity in Koopman and Matza for the sake of greater accuracy.

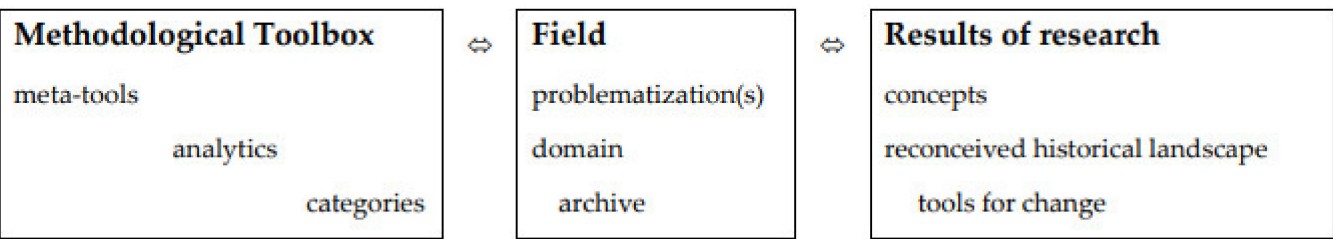

**Figure 2.** My revised Foucauldian framework.

This schema helps situate the rest of this paper as an attempt at (flexibly) practicing Foucault at the problematization—or topic—of CSR, as located within the broader topic of the corporation. One might quite naturally think of 'the corporation' as a field, like one might think of 'the clinic' or 'sexuality' as a field, which is fine shorthand, and which I will do in this paper. According to my schema, the full specification of a field would also

include specifying the 'domain' and 'archive'. 'Domain' is meant to provide a historical, temporal-spatial context, and 'archive' is meant to refer to the available discursive remnants through which we access that object in its domain.

I should note that "problematization" is a fascinating (though perhaps not fully developed) idea which emerged in the late Foucault: a problematization is a historical "totality of discursive or non-discursive practices that introduces something into the play of the true and false and constitutes it as an object for thought" [10] (p. 257). However, while madness and crime clearly count as problematizations [11], it is not so clear whether institutions such as the clinic or the prison count as problematizations. Additionally, Foucault was less explicit about the researcher's own problematization of their field—e.g., the manner in which Foucault was problematizing madness, or problematizing the historical problematization of madness. So, while it is unclear whether 'the corporation' would count as a problematization, CSR appears as a very suitable candidate, since it arises in the context of a need, or problem, of evaluating corporations and giving true (or false) assertions about them. However, having at least flagged this issue, I will not overly concern myself with what exactly counts as problematization. Instead, I loosely employ the term 'problematic object' or simply 'problematic', to cover any concept or institution which both arose in connection with a historical problem and which the researcher (Foucault, myself, or others) takes up as a target of critical study.

Thus, we can 'try out' the idea that CSR is a 'problematic object' of analysis, and a 'subfield' of analysis, within the broader field (and problematic) of the corporation. In the next three sections, I will embark on a methodologically and conceptually Foucauldian approach to this subfield of CSR in connection with the broader field of the corporation. I will start with an introductory discussion concerning the umbrella field of the corporation.

## 3. A Ripe Field for Foucauldian Analysis

Foucault was a master at finding fields of research which were ripe for analysis, reflection, critique, and 'think[ing] otherwise' [5] (p. 327). Although his research often comes across as a rupture—as a new event, or a potent discourse which cuts across and disturbs old discourses—it typically does not do this by creating a new field from scratch. It usually does this by taking an old field and turning it upside down, or by shining lights into its dark corners, or by merging together the boundaries of several old fields in order to create a new one. This is what is needed in order to examine not medical history, but the birth of and discourse surrounding the clinic; not political history, but the governmentality manifested in prisons and 'reasons of state' and neoliberalism; not ancient ethics, but the modes of subjectivity carried out in sexuality and 'care of the self'. As hinted at in the last section, he typically does this by finding a 'problematization', or a problematization within an institution, or several related problematic objects. For instance, 'madness' was a problematization behind the 'the clinic', behind 'the prison' was 'crime' and these were connected to the problematic of 'punishment,' and 'biopolitics' was connected to public health, 'security', and 'neoliberalism' (see [12] (p. 120) for a depiction of 'going behind' institutions). There is not the firmest of divisions between problematizations, my looser conception of a 'problematic', and the concepts Foucault develops through his research. For example, arguably 'care of the self' could fit in any of these categories: a totality-problematization, a vague problematic object, or a concept emerging from research, in *Hermeneutics of the Subject* [13].

In the 1970s, Foucault found the state to be a sort of broad ripe field for this sort of thing—for turning it upside down and unearthing its dark corners. Political history and political philosophy were well-trodden fields with ancient grooves of thought, and the state had become a theoretical object for political theorizing, but Foucault had a way of taking a favored object and critically upending it, so as to make it into a historical–empirical object ready for excavation. Theoretical Marxism's manner of taking up the state as a research object might be described as critical but not empirical; it tended to regard the state in an abstract or even transcendental manner, e.g., Louis Althusser asserted that "The whole

of the political class struggle revolves around the State" [14] (p. 140). It is hard to avoid the temptation to infer that Althusser was doing a sort of 'reifying' of the state as "the State"—as a powerful agent. (It is also worth considering whether the more recent *Empire* (Michael Hardt and Antonio Negri) [15] reifies "Empire" similarly.) Althusser's student, Foucault, would take up the state as a field of research, but in a different manner: as a broad historical research field. The state was a field rich enough for Foucault to find a variety of problematizations, institutions, and concepts within and connected to it, e.g., 'madness' and 'crime' were problematizations occurring around and within the state; clinics and prisons were gradually taken up and authorized by the state; and punishment, discipline, security, surveillance, governmentality, and biopolitics were Foucauldian 'concepts' emerging from his encounter with the field of research. Some of these concepts would be described as 'technologies' or 'dispositives'. ('Dispositive' means roughly a social pattern imbued with power. I will discuss the 'dispositive' concept in more detail later in the article.) A key research move which Foucault made as he developed these concepts otherwise than Marxists was to decenter and demythologize the state [16], such that the apparatuses (or dispositives) at play did not emanate from the state (as Althusser had depicted), but were dispersed in society in a decentralized manner (see Bernard Harcourt's discussion in [17] (pp. 265–310)). Through these ways of examining the state, Foucault joined Althusser in undermining old grooves of thought which conditioned political choice in limited ways, such as framing choice as being between living inside or outside the law, or between respecting or deriding a national leader, or between joining one of several mainstream political choices. Marxism had opened up new choices, such as choosing whether to challenge economic systems and ideologies. But Foucault's way of thinking otherwise opened up ways of conducting 'resistance' or 'counter-conduct' in more diverse and tactical ways than the grand ideological and revolutionary choices opened up by Marxism.

In order to think about a Foucauldian approach to the corporation, it is helpful to first reflect more broadly on how Foucault's methodologies developed in relation to Althusser and Marxism. I interpret Foucault's reaction to Marxism as complex and as involving highly important methodological differences (in which aspect I disagree with Jeffrey Nealon, who tries to 'link' Foucault to the neo-Marxist Fredric Jameson [18] (p. 56, and Chp. 3)). On a broad level, Foucault became suspicious of grand or totalizing 'theory': "The role for theory today seems to me to be just this: not to formulate the global systematic theory which holds everything in place, but to analyze the specificity of mechanisms of power" [1] (p. 145). In direct response to "a faded Marxism", Foucault advocated encountering "problems which are specific, 'non-universal', and often different from those of the proletariat or the masses" (p. 126). I interpret this as putting Foucault as importantly different from the Marxist tendency to give a grand, systematic, and roughly economic diagnosis of society—whether that diagnosis centers around the 'bourgeoisie', the 'State', 'capitalism' (Jameson) [19], 'control' (Gilles Deleuze) [20], or 'Empire' (Hardt and Negri) [15]. Arguably, this allows Foucault to circumvent the danger of evacuating all power and agency from the level of the subject in order to assign it to the highest social-system level (a concern in [19] (p. 5)). Foucault conceives of power as dispersed throughout and analyzable within various societal levels and angles, including institutions, practices, discourses, individuals, and "sub-individuals" (see [1] (p. 208) where Foucault again directly opposes Marxist theory). This also allows resistance to occur at various levels, rather than just at the level of a class (or multitude). Thus, in various ways, Foucault thinks otherwise than both Marxism-supporting methodological collectivisms and capitalism-supporting methodological individualisms (see Joseph Heath's article [21] as a starting point regarding the amenability of methodological individualism to Austrian School economics).

However, one could definitely make the case that Foucault overreacted to Marxism, or that he could be a worthy ally of non-methodologically collectivist types of neo-Marxism, or that he shied away from the economic, or that his occasional remarks in recognition of economic classes or discourses were not fully investigated in his books and lectures (*The Birth of Biopolitics* [22] being the main exception). He showed far more attention to

aspects of the state than to aspects of the economy and the economic. Thus, for multiple reasons, I believe that the corporation has become a ripe field for Foucauldian analysis—even overripe. Not only did Foucault neglect it, but many types of Marxist theorizing (such as those mentioned above) tend to look above the corporation to a broader systemic order, and systemic narratives can project universality upon messy empirical details (as suggested by Jonas Hagmann) [23] (pp. 435–436). The corporation occupies a significant position insofar as it is the main institution within the powerful realm of the economic. This institutional position is comparable to that of Foucault's targeted institutions such as the clinic, the madhouse, or the prison—all these institutions are sites in which practices and strategies coalesce and involve power, discourse, and subjectivity—it is exactly the kind of problematic object, or field, which calls for a Foucauldian investigation. For just a taste of the many resources supporting these ideas and illustrating some ways in which the Foucauldian investigation I am calling for has already begun, see Gilles Deleuze [20] (p. 6) on the corporation as an institution of power, Paul Thompson and Stephen Ackroyd's [24] discussion of strategies of corporate subjectivization of labor ('humanistic control', p. 620), and Ron Wagler's [25] analysis of the technology and subjectivization of customers through advertising (arguably a new sort of discourse). In a Foucauldian manner, this article intends to make further headway into looking within and behind the corporation, and to ask whether our Foucauldian toolbox is properly equipped for the task.

Although I am deliberately creating distance between my Foucauldian approach and many Marxist approaches, I sympathize with the sense of urgency often found in Marxist thought—the sense that something must change, perhaps drastically change, if one is to avoid becoming subjectivized and shaped as a pawn in all sorts of new ways by corporations and broader economic dynamics. It can seem like one's lifeworld is threatened from every side: all day long one uses the devices and software and products produced by corporations, sees scatterings of corporate logos across every screen and container, one does one's best to avoid getting distracted by corporate ads while producing for corporations and consuming corporate products, and one tries to relax to entertainment produced by some corporations provided on the website interfaces of other corporations. For many, "the state" fades relative to the corporation as a dual influence on their subjectivity as both employees [24] and consumers [25]: their encounters with the political and with the actions or policies of their government, largely provided in media managed or owned by corporations, becomes much more contingent compared to the new 'historical a priori' of transacting with (and within) the products, processes, and traces of corporations, resulting in all sorts of visible and invisible deposits on their subjectivity. There is an intensification of the economic [18] (Chp. 3), including a public-to-private shift (surveyed in [23] (pp. 420–421)). There is reason to ponder how the Althusserian conception of interpellation (meaning roughly the recognition and constitution of subjects via a ruling ideology) might be usefully privatized (rather than seen as a state affair): how do corporations and markets interpellate subjects, such as through advertising [25] or by digitizing them (as argued by Richard Weiskopf) [26]?

However, a counterpoint to the last paragraph is the thought that corporations are themselves entangled in a big social web, since corporate structures and actions are regulated by various governmental agencies, which are regulated by legislation voted on by politicians, who are lobbied by corporations. Not to mention the cycles of interactions between corporations and consumers, corporations and employees, governments and citizens, etc. There is a messy network of 'actors', but not all nodes act equally or act consciously. Also, corporations may be a sort of recent historically emergent 'actor', but a complex network lies inside such an 'actor'! The corporation has a disputed, vague identity, which gives off the appearance of being a 'private' entity within the 'private' sector, but which is better thought of as a hierarchical organization which is not entirely private or public, David Ciepley suggests [27]. Given the various power dynamics and strategies entangling all these nodes, Foucault's concept of 'conduct of conduct' becomes quite relevant. There are many types and ways of conducting conduct, as well as ripples describable as 'conduct

of conduct of conduct', 'conduct of conduct of conduct of conduct', etc., as well as joint conducting of conduct, misinterpreted conducting of conduct, and various permutations of misconduct. Rather than represent the whole as a unity or a hegemony, Foucault described the broad power landscape as more of a site of dynamic struggles and resistances—a "civil war" [17] (p. 229). Later (Section 6), I will suggest that the dynamics of this "civil war" are not entirely captured by Foucault's surface analytic which claims that everything is "always said," as Foucault declares in *The Punitive Society* [17] (p. 37) (also stressed by Deleuze [28]).

The play of forces or power—the causal 'physics'—in which corporations make decisions, establish practices, and interact appears contingent and uncontrolled. Commensurate with Foucault's oppositions to grand theories, Foucault opposed the search for a single foundational cause as well as the idea of causal determinism [29] (p. 5). Rather, he mobilized Althusser's idea of "overdetermination" [1] (p. 195)—i.e., although a single cause could potentially function so as to create an effect, social situations (such as practices, institutions, and events) are often (or always) places where multiple causes tend to struggle to effect and determine. Kaspar Villadsen summarizes: "more causes are present than what is necessary to cause the effect" [30] (p. 5). This opens up the exploration of a network or intersection of causes, forces, strategies, and 'dispositives' at play in any corporate development, as well as in the whole economic sphere, or in society more broadly. This seems like an apt approach to a complex reality in which one can hardly conceive of a boardroom decision or product design driven by a single cause. Such events tend to happen at an intersecting overflow of patterns of thought and practice from various spheres, such as the economic, the state, and other aspects of society.

In this section, I have attempted to lay a basic Foucauldian framework for approaching the corporation as a research field, I have distinguished such a project from related Marxist endeavors, I have given some motivation for engaging in such a project, and I have offered a glimpse of the array of issues surrounding the corporation field and the CSR subfield. In the next section, I will use, and in a way, sharpen, the Foucauldian tool of the 'dispositive' in order to provide more specificity to the Foucauldian approach to the corporation which this section outlines in broad strokes. While this section has been more concerned with methodology, the next section will be more concerned with concepts.

## 4. The Corporation as a Field of Play of Dispositives

Just as Foucault found dispositives to be helpful concepts for understanding the workings of power within institutions such as the clinic and the prison, this section will utilize Foucault's notion of the dispositive to help us to understand how power operates in the corporation. However, I will also argue that Foucault's tool of the 'dispositive' was underdeveloped. A fuller picture of how dispositives can be analyzed in a research field will better equip us to apprehend the power dynamics at play in a corporation and corporation-related discourse. Foucault's often-cited depiction of the dispositive is that it is a thoroughly heterogeneous ensemble consisting of discourses, institutions, architectural forms, regulatory decisions, laws, administrative measures, scientific statements, philosophical, moral and philanthropic propositions—in short, the said as much as the unsaid. Such are the elements of the [dispositive]. The [dispositive] itself is the system of relations that can be established between these elements [1] (p. 194).

Although that comment was preceded by a reference to a dispositive of sexuality, it seems that the most discussed examples of Foucault's dispositives are law, discipline, and security (e.g., by Villadsen, and by Sverre Raffnsøe et al.) [30,31]. However, "any schematic representation of [Foucault's] dispositional analysis entails both interpretation and reconstruction" [31] (p. 281). Deleuze's interpretation [32] elaborated on how a dispositive includes aspects of knowledge, power, and subjectivity, and emphasized the multiplicity of the dispositive. However, one must also ponder what unites a dispositive into a single term. A dispositive is not just any old ensemble or multiplicity, it is a network of connected elements—broadly speaking, a type of *pattern*—a societal pattern imbued with

power. So perhaps a fuller account of a dispositive is that it is a pattern of disposing and being disposed which is spread across a "heterogeneous ensemble" of societal elements and which intertwines aspects of knowledge (or discourse), power, and subjectivity.

However, I wish to make a Foucauldian point about the 'dispositive' itself. My claim is that Foucault tended to look for patterns of power with a particular sort of breadth. This breadth was not supposed to characterize all of society in a grand or determinate way—it was not supposed to be as thoroughly dominant as Althusser's 'State' [14], Jameson's 'capitalism' [19], or Hardt and Negri's 'Empire' [15]. Nor was it supposed to be as deeply engrained into the background of society as the broader categories of 'power,' 'discourse', or the 'subject'. On the other hand, it should be more extensive than more particular 'strategies': a strategy such as surveillance was apparently not quite broad enough to be mentioned as a dispositive, nor a manner of punishment such as the public spectacle, nor (presumably) an approach to care-of-the-self offered by any one philosophical school. There is one counterexample (mentioned by a reviewer), which is that Foucault did make an early reference to the asylum as a dispositive [33], which seems to have a different scope than Foucault's broader examples. For Foucault's main examples, there is a goldilocks quality to the dispositive: it should not be too broad, nor too specific—it should be an analytical pattern which is at a level of analysis which is "just right".

This terminological issue—the issue of the proper 'scope' or definition of a Foucauldian term—came up before, in that there is a similar difficulty about what properly qualifies as a 'problematization'. Likewise, there is a similar terminological issue with the term 'governmentality'. (Perhaps this sort of issue is a downside to a non-systematic thinker like Foucault.) However, there is perhaps a difference of note regarding the term 'governmentality'. Foucault's conception of governmentality began as a more specific term which he applied to designate political governance as "the art of governing" (a common phrase in *Security, Territory, Population* [12]). However, he then broadened it to the more comprehensive conduct of conduct [34] (p. 221), which could operate at various levels (micro or macro, social or political).

What I want to propose is that we consider doing for Foucault's 'dispositive' what Foucault did (or did more clearly) for his 'governmentality'—namely, allow it to span different scopes, rather than use it to try to nail down patterns at some particular goldilocks level of analysis. Whether or not one agrees that this would be terminologically helpful, one should at least be open to the possibility that patterns (and assemblages) of power occur with varying degrees of breadth and over various spans of time. This means that patterns of lesser breadth could be thought of as local or regional dispositives, and that while some dispositives span or even refer to a single institution (e.g., 'asylum dispositive'), others span multiple institutions (e.g., 'discipline'), and still others are somewhat non-institutional (e.g., 'sexuality'). Additionally, one should be open to the idea that the breadth of a dispositive might not precisely reveal the strength or nature of its power. For instance, the asylum, the barracks, and the school system could all be thought of as dispositives of comparable breadth, though they have different contours and strategies of power. Finally, we should imagine dispositives as entangling in a variety of ways—at times supporting or allying with each other, at other times competing or warring with each other. For instance, discipline and security might clash, they might clash in different ways inside different institutions, and they might clash with dispositives of other breadth (such as local dispositives) which are at play within a particular institution.

What this means for the corporation is that we should expect the corporation to be a site in which various dispositives are entangled—dispositives of various breadth which operate along different contours of power. For instance, it is worthwhile considering whether there are specifically economic dispositives which are quite broad but have a different quality than security and discipline: patterns such as profit-seeking, 'economic rationality,' marketization, and measurability. Other dispositives operating within corporations might be more regional or might be better conceived of as strategies of broader dispositives: these include surveillance (Foucault) [35], networking (Weiskopf and Iain Munro) [36], human

capitalization [36], digitization (Weiskopf) [26], standardization (Laura Silva-Casteñada and Nathalie Trussart; Elena Escrig-Olmedo et. al.) [37,38], and flexibility (Chris Benner) [39]. Security and discipline are an example of dispositives which can compete, such as in their approaches to risk, whereas digitization and standardization are an example of dispositives which make for good allies.

Recall that Foucault's concept of overdetermination allows for a play of different forces, including forces at different levels (group, institution, individual, and sub-individual). A broadened conception of dispositives gives more color to this: I suggest (with Raffnsøe et al. [31]) that the concept of the dispositive helps elaborate Foucault's overdeterminism. This is because the clashing of dispositives involves a clashing of forces attempting to determine situations and subjects.

One upshot of my perspective here is that although the dispositive has been explored beyond Foucault's initial depictions, it continues to carry more potential for conceptual employment and experimentation. When we start to open up a dispositive research space between the (helpful but limited) trichotomy of law–discipline–security and the unlimited Deleuzian ocean of "multilinear ensemble[s]" [32] (p. 159), then we open up an ensemble of good research questions. Does a concept such as "empire" or "control" have enough unity as a pattern to qualify as a dispositive? How do dispositives interact, such as security and economization, or human capitalization and surveillance, or flexibility and measurement? How do broad dispositives become localized in certain fields—such as, can we think of accounting as a local form of a broader dispositive of measurement? How do dispositives evolve within new contexts, such as when security and surveillance are fed big data [26]? How can it be argued that one dispositive (e.g., 'control' or 'security') is pre-eminent (an idea Foucault touched upon [12] (p. 144)) in relation to others? There are many potentially fruitful directions of enquiry about both dispositives at play in the corporation and dispositives at play in broader society. In short, if we think about Foucault's toolbox in a flexible way, we have a very useful and open analytic for engaging in a field of research such as the corporation. Namely, a corporation is an institution in which a collection of various dispositives are at play in both cooperative and competitive manners—in which some patterns serve as strategies for broader patterns, and other patterns can be viewed as localized forms of broader patterns.

Finally, although I have highlighted the power aspect of dispositives, the discursive aspect is highly significant. Discourse is not only a dimension of the dispositive's operations—part of a dispositive's 'ensemble'—but is also a focal site of struggle, including struggle about a dispositive, against a dispositive, or between competing dispositives. This struggle can be particularly intense in regard to claims about truth and knowledge which pertain to the setting of norms and ethical principles. This is why the ethical discourse surrounding corporations carries important power effects. It is within that normative discourse, in which dispositives are engaged in a power struggle, that the importance of CSR appears.

Furthermore, there is an important sense in which a broader conception of the dispositive helps open up conceptions of the discursive aspect of the dispositive. Most of Foucault's comments about dispositives came from before his ethical turn, especially from the 1970s when he was concerned with the entwinement of power and knowledge, and in that context the discursive aspect of a dispositive was characterized as a formation or production of knowledge [30,31]. However—here I am foreshadowing my argument in Section 6—just as not all dispositives operate along parallel grooves of power, I suggest that not all dispositives operate via an articulated knowledge formation. This is particularly relevant if (as I will argue in Section 6) not all discourse can be adequately analyzed as knowledge oriented.

## 5. CSR as a Discursive Event and (Sub)Field of Discourse

CSR has been described in several ways: as "a social and political phenomenon in the broadest sense" (Steen Vallentin and David Murillo) [40] (p. 825), as "a field of action"



(Ronen Shamir) [41] (p. 669), as a "definitional construct" (Archie Carroll) [42] (p. 268), and (roughly) as a field of discourse (Kristian Quistgaard Steensen & Villadsen) [43]. I will not give a comprehensive history of CSR, but I will give some arguments regarding the history of CSR which has led to its present situation, keeping in mind my method of using Foucault's tools flexibly and hermeneutically.

The historical event of new names and terminology comes in different textures. Sometimes such names appear in theoretical discourse highly abstracted from practices, such as when Deleuze, Martin Heidegger, or Foucault assemble new terminology to form new theories or perspectives. Sometimes names are created in order to initiate new actions or practices, such as when a law is proposed and passed around the legislature and the news. Sometimes names are retrospective ways to refer to events or practices which have already happened or begun, such as when a war, revolution, or international incident gets named.

In the case of CSR, there was no special event which needed naming—no war or revolution or even more mundane observation along the lines of "wow, this company has an amazing way of doing things, let's name this particular method 'CSR'". Rather, CSR started out as primarily a discursive event, originating with the publication of Howard R. Bowen's book *The Social Responsibilities of the Businessman* in 1953 [44].

What preceded the term 'CSR' was the increase in big businesses and corporations in and on the wake of the second industrial revolution—"The Rise of Big Business", as Glenn Porter's book calls it [45]. Similar to how the first industrial revolution had created opportunities and needs for businesses to utilize early forms of mass production, businesses had been increasing in "Scale and Scope" (as Alfred Chandler's book elaborates [46]) and utilizing new forms and strategies of business organization to manage various increases in scale and scope. This included the "separation of ownership from management" (p. 1). The challenges facing the corporation and its social integration created a variety of needs—legal, civic, and educational. There arose satellite institutions such as "trade associations, better business bureaus, civil society organizations, business schools, [and] business ethics associations" (Gabriel Abend [47] (p. 28)).

In light of what may be called an 'ensemble' of heterogeneous elements which were coming together to support the institution of the modern corporation, one could argue that the corporation itself is a sort of dispositive. However, in my opinion, a better candidate for a deeper pattern or dispositive is that of what I call "consolidation". What I mean by consolidation is the pattern of expansion, integration, and combination (merging and acquiring). I suggest that consolidation is a broad, nonlinear pattern with different features during different industrial revolutions, in which family businesses were pulled towards becoming small businesses, small businesses were pulled towards becoming or being swallowed by large businesses, large businesses were pulled towards becoming or being swallowed by corporations, and large businesses and corporations were pulled towards becoming international or global. One could also consider consolidation to be a strategy of the broader dispositive of profit seeking, but one could also imagine other factors (besides profit seeking) which drove consolidation, such as survival, ambition, and sometimes even nationalism. Institutional consolidation involved power consolidation, and that sort of power—the power which was increasingly concentrated in the corporation—created a need for a discursive shift to help contest that power.

The satellite institutions mentioned had already been host to a conceptual and discursive shift which had arisen with (and following) the rise of big business in the second industrial revolution. That shift, as revealed by Abend [47], included a shift in what he calls the "moral background"—the conceptual framework on which moral issues appear. In particular, the "object of evaluation" (p. 92)—the object which can be treated as a moral agent responsible for its actions and which is worth attention and moral evaluation—was shifting.

Ethical discourse was shifting from ascribing agency and responsibility to the individual— the merchant, businessperson, or customer (Benjamin Franklin comes to mind)—to ascribing agency and responsibility to the business or the corporation. Agency and responsibility were also extended to "business" or "American business" conceived as a collective

entity [47] (p. 50)—a development which resembles the formation of later terms such as "corporate interests" or "big business". One might say that there was a consolidation of agency in connection with the consolidation of businesses.

This shift in consolidation, agency, responsibility, and discourse—from individuals such as Ben Franklin's moralized reader and the "Christian merchant" [47] (p. 633) to the broader targets of business ethics, corporate ethics, and the corporation—was accompanied by a shift in discursive spaces. Past discursive spaces included the sermon, didactic books, and the local newspaper. However, seeing as businesses (unlike individual merchants and businesspersons) were not subject to conversion, repentance, or religious 'pastoral power' [12], there arose a vacancy—a need for new discursive spaces and concepts. To some extent, I believe this connects with an extrapolation or 'consolidation' of the Christian message into a more social message—the 'social gospel'—which Steensen and Villadsen argue influenced the original development of CSR [43]. But what also emerged was the shift to the classroom and to academic scholars and scholarship—the need for classes such as business ethics and concepts appropriate to those classes.

It was in the context of institutional consolidation, agential consolidation, and discursive space shift that the term CSR was needed and welcomed. I do not believe that CSR arose as a radical call to action [43], or at the other extreme, as a maneuver of corporate interests to claim that they were responsible so as to resist the need for the state to regulate them. Rather, I suggest that the creation of the term 'CSR' was an overdetermined discursive event—an effect of consolidation and corresponding shifting ascriptions of agency, a part of the transition from economic normalizing discourse into classrooms and scholarship, a part of a broader trend toward specialization and expertise in academia and beyond, a response to the business consolidation which had accelerated during the second industrial revolution, a new context for Christian values to interpret themselves, and an effect of deeper societal trends of economization and economic rationality and of resistance toward those trends.

Since the processes leading to the creation of CSR would only intensify as the 20th century progressed, CSR became not just a term or a discursive event, but its own discursive zone—a discursive subfield lying at the intersection of discursive fields of ethics, economic philosophy, business, and the corporation. CSR became a site for power struggle—for battling out terms and trends and dispositives, for establishing resistances and counters to those resistances, and for advancing or defining radical change and opposing it.

## 6. CSR Discourse and an Insufficient Toolbox

This section marks a shift in the direction of this paper. Whereas the preceding sections mainly focused on applying Foucault's toolbox to the corporation and CSR, this section attempts to assess the current nature of CSR discourse "on its own terms" in a manner which calls into question the sufficiency of Foucault's toolbox to properly assess CSR discourse. The general reason for this is the vast difference between the nature of CSR discourse today and the discourse which Foucault researched.

I call the discursive fields which Foucault researched in his lectures and books, excluding the more broadly oriented *The Order of Things* and *The Archeology of Knowledge*, his "classic" research fields (or "classic discourses")—these include the discourses surrounding the clinic, madness, abnormality, the prison, discipline, biopolitics, and ancient ethics. Although knowledge/truth/discourse could be considered a single axis in Foucault, I noted above (Section 2) that while 'truth' was a more helpful lens for Foucault during his 'ethical' phase, 'knowledge' was more helpful in much of his earlier work. In his 1982 lectures *Hermeneutics of the Subject* [13], access to 'truth' is typically a personal accomplishment earned through personal transformation, and its circulation tends to be limited to mentorship or teaching accompanied by various practices. 'Knowledge' is a more useful lens for capturing the nature of accepted statements in modernity, and 'knowledge' followed certain institutionalized rules of formation, circulation, and archiving. I propose

that 'discourse' is the umbrella term here: some discourse is largely knowledge-oriented, and other discourse is largely truth-oriented.

My central claim in this section is that *the discursive field of CSR is neither knowledge-oriented nor truth-oriented*—it differs substantially from Foucault's classic discourses. The import of my claim is that Foucault's toolbox for approaching discourse in terms of knowledge or truth is insufficient. I will elaborate two broad arguments for my claim: the nature of CSR discourse differs from that of Foucault's classic discourses, and CSR discourse circulates differently from that of Foucault's classic discourses.

To start with, I claim that the *nature of CSR discourse* is different than that of Foucault's classic discourses of study. It differs with respect to the situation of the speaker or source of discourse, the nature of the archive, and manner in which CSR discourse circulates.

First, the situation of the speaker, or source of discourse, is different. In knowledge-oriented discourses, Foucault emphasized the statement and its institutional and archival context—in other words, the discursive position of statements—while de-emphasizing the speaker, subject, or 'man' issuing the statement. This methodological background to discourse was hinted at in *The Order of Things* and given expression in *The Archeology of Knowledge* [6,48]. In truth-oriented discourse, the situation is different—in *Hermeneutics of the Subject* [13], the speaker, listener, and the speaker-listener relationship (such as teaching, mentoring, or guiding) is a highly relevant part of the discursive situation—it gives the context in which *parrhesia* or other modes of speech can appear. As for CSR discourse, the speaker's identity, perspective, and reputation comprise an important part of the discursive situation. The source of a CSR claim—whether that is Greenpeace, a corporate website, or a particular news outlet—is highly integrated into the claim's situatedness in discourse. Furthermore, there are often issues of murkiness surrounding the interests and/or identity of the CSR speaker. Independent contractors, former executives, family members of employees, and corporate bondholders are some examples of parties whose interests may or may not be aligned with the interests of the company or of its shareholders, because such parties are not entirely situated on the company's interior or exterior. It is not a simple black-or-white matter of whether the speaker is strictly inside or outside the company or whether the speaker's interests are aligned with or against that of the company. Thus, the situatedness of the speaker is itself a matter of potential interpretation. In light of these observations, firstly, there is a stark contrast between CSR discourse and knowledge-oriented discourse. The relevance of the speaker for CSR is a marked contrast from the subsidence of the speaker in Foucault's approach to knowledge-oriented discourse. Secondly, the comparison between CSR discourse and truth-oriented discourse is a little closer, because the speaker is relevant to both. However, the issues surrounding the speaker in CSR discourse, such as murkiness and perspective, are different than that of truth-oriented discourse, which include the relationship between the speaker and listener. This difference between CSR discourse and truth-oriented discourse may be thought of as a contrast of connection—truth connects and enhances connections between speaker and listener, whereas CSR discourse lies in a collection of mediated spaces between a presumably disconnected source and audience where any personal connections are accidental.

Second, the 'archive' of CSR discourse differs greatly from that of Foucault's classic discourses. I suggest that Foucault's theory of archive as articulated in *The Archeology of Knowledge* [6] arose as a reflection upon reflections of modernity—a sort of doubly removed apprehension which nevertheless maintained contact with modernity in that *knowledge* remained a basic analytical contact point as the archival object. This allowed Foucault to problematize history, continuity, and the subject, but underlying those problematizations were certain stabilities, such as the document, knowledge, and the memory aspect of the archive (documents become "monuments", p. 7). However, I suggest that what we see now, both in corporation discourse and other discourses, is that there is no longer a stable document, a shared archival memory, or a stable underlying contact point of knowledge—we have a different archival situation. Instead of the 'document', we have an assemblage of

new and old media, including an aggressively expanded utilization of advertisement and advertising tactics (Wagler; Sebastião Vieira de Freitas Netto et al. [25,48]. Instead of only a reliable, shared, centralized archival memory (represented by the library), we have both a multi-valanced shared memory (e.g., library and internet) and a fragmented, contested, and sometimes politicized private memory (including one's email history, one's social media history, and one's records of purchases) (Jacques Derrida raised this issue of contrast [49]). The archival memory of truth is hardly attended to by Foucault, but in *Hermeneutics of the Subject* [13], it can be thought of as collections of philosophical documents and sayings which were supported by auxiliary collections of letters and journals. In contrast, for a contested discourse such as CSR, instead of an archival object of knowledge or truth collected in documents, what is archived consists of a collection of competing presentations, perspectives, and interpretations collected in various media. There may still be a sense in which CSR discourse orbits around knowledge, but that orbit is subject to mediation, commercialization, and interference in ways that Foucault's classic discourses were not.

Finally, CSR discourse does not fit into the grooves of knowledge-discourse or truth-discourse—it circulates differently. When a company says "our product does X", "we here at X value the environment", or "we apologize for our oil spill and hold employee Z responsible", these statements are not subject to the copy-and-transmit mechanism of knowledge, or the practices guarding the speaking and hearing of truth (discussed in *Hermeneutics of the Subject* [13]). Rather, they are circulated unguarded and at a step removed: e.g., "On Thursday, company Y reported 'our product does X'". There is a space, or spaces, between a statement or presentation of a company and straightforward knowledge—including the space between the statement and the way it is received. This space was smaller and simpler in Foucault's classic discourses—it was typically built upon dichotomies, perhaps a space of several dichotomies: of true/false, acceptance/rejection, and scientific/non-scientific. The new space is inhabited by questions such as: "How committed is the company to this statement? What is the hidden intention behind this statement? Is the statement intentionally vague to hide specific details or a lack of specific details? Is the statement intentionally specific so as to hide a general tendency? Who is making the statement, and what do they want?" Even subregions of discourse with strong aspirations for objectivity, such as accounting (Mario Abela) [50] and third-party certification (Escrig-Olmedo et. al.) [38], become subject to explicit contestation and a-step-removed circulation when they enter the arena of CSR discourse. A subregion of 'discourse' with quite different aspirations is the non-verbal symbolic impressions of a company which circulate in a disseminated manner through advertisements and websites. This is also not the circulation of knowledge or truth, which, by contrast, are suited for conscious, articulated, and individualized circulation.

Next, I claim that another sign that corporation discourse (including CSR) does not fit aptly into Foucault's knowledge or truth framework is that it does not merely differ in nature, but it also *subjectivizes* differently than Foucault's classic discourses of study. In *The Birth of Biopolitics* [22], Foucault's application of his old toolbox to neoliberalism led him to depict a new *homo eoconomicus* apprehension—arguably, a subjectivization—in a manner which was quite limited by his straightforward approach to discourse as knowledge-oriented or truth-oriented. He connects the formation of *homo eoconomicus* to a Humean "subject of interest", in which there is "the appearance of interest for the first time as a form of both immediately and absolutely subjective will" (p. 273), leading to a *homo eoconomicus* which is "a sort of non-substitutable and irreducible atom of interest" (P. 291). However, he appears aware that he is merely tracing the theoretical, knowledge-oriented discourse of neo-liberal thought, rather than attending to the actual experiences of human beings: "*Homo eoconomicus* is, if you like, the abstract, ideal, purely economic point that inhabits the dense, full, and complex reality of civil society" (p. 296). My claim is that although Foucault mentions this 'complex reality', he is not well-equipped to assess it. His categories of knowledge and truth are insufficient tools to assess the multivalent desire-toyed subjective situation of, e.g., the sort of customer who is pulled by the desire to save

money for their family, the lure of the catchy music in the somewhat cliché ad which they had to endure before watching a YouTube video (an ad which appeared because of some algorithmic advertising decision [26]), and the half-hearted attempt to be a responsible consumer by trying to recall an accusatory article which they skimmed about this company last year, and by perhaps starting to research the company's responsibility and reputation (which they feel they hardly have time for, and which soon gets interrupted). The contrast between Foucault's approach to subjectification in his classic fields, and the complications of subjectivization tied to corporation discourse, provide another reason for reconceiving and renovating Foucault's approach to discourse.

If Foucault's approach to discourse as knowledge oriented or truth oriented is not apt for approaching CSR discourse, the alternative which I am suggesting is that CSR discourse is subject, at least to an extent, to decipherment and interpretation. In other words, I am suggesting we consider in what ways a statement might be more than what it says explicitly, including the possibility that it may have a sort of 'below' or 'behind' the surface aspect. What I am suggesting is that we give up what could be termed Foucault's "positivity meta-tool", which Deleuze summarized as follows: "That everything is always said in every age is perhaps Foucault's greatest historical principle: behind the curtain there is nothing to see" [28] (p. 54). (I will discuss this more in the next section.)

My term for the space which surrounds a 'statement'—which lies behind or beneath it, and into which one seeks for un-explicit context and meaning—is 'hermeneutic space'. However, it seems that the plural, 'hermeneutic spaces', is more apt, since there are a variety of potential gaps, or distances, surrounding a statement. These distances include the distance between a deposited statement and its original intended meaning and context, between the context in which a statement is received and its original context, between the speaker's intended meaning and the speaker's intention for the statement's effect, between the speaker's intended effect and the actual effect, between the intended effect and the audience's decipherment of the speaker's motives and intended effect, between the real and interpreted identity of the speaker, and between the original meaning and various interpreted meanings (J. L. Austin [51] is helpful in discerning some of these distinctions). In order to make sense of hermeneutic spaces, I believe that we need some sort of hermeneutic tool(s).

One might ask the following two questions about hermeneutic spaces. First, isn't there always a distance between a speaker and recipient, or between an author and their deposited word and the uptake of that intended word—how is CSR different than other regions of discourse? Secondly, what kind of implications does a recognition of hermeneutic spaces have on the dynamics of discourse and power in the CSR field? In the next section, I will consider strands of hermeneutic thought more broadly in order to suggest specific ways we can renovate Foucault's toolbox so as to be equipped to tackle the first question. After that, in Section 8, I will turn back to the specifics of the CSR field and exemplify how such a renovated toolbox can give us an enhanced picture of CSR discourse and its power dynamics.

## 7. Hermeneutic Theories and Tools

Historically, there were two regions of hermeneutical development, which might be termed 'positive' and 'negative'. The positive kind had the shape of a trajectory or tree: it started based in Biblical interpretation, then gradually broadened to first address interpretation of texts more generally (Friedrich Schleiermacher), and then to encompass general understanding of the humanities or of human experience (Wilhelm Dilthey, Heidegger, Hans-Georg Gadamer, Paul Ricœur, etc.). The 'negative' hermeneutics to which I am referring, namely the hermeneutics of "suspicion," has less trajectory or structure. They are a collection—a small collection of diverse approaches of 'unmasking' to reveal a darker origination beneath a presented surface. More specifically, Karl Marx unmasked class struggle beneath the surface of social-economic activities, Freud unmasked psychodynamics beneath the surface of the psyche, and Friedrich Nietzsche unmasked instinct and

power beneath the surface of morality and society. (Here I am roughly following ideas from Ricoeur (2008), who collected the negative hermeneuticists under the term "suspicion.")

If one takes either of these regions of hermeneutics to their broadest limit, then one may end up at a point of contact with the idea that all we have is interpretation— there is nothing, or nothing accessible, outside of interpretation. One might call this 'epistemological hermeneuticism'. Nietzsche was most conscious of this idea and seemed at times to advocate it (this is especially evident in *Will to Power* [52]–see sections 15, 477, 481, 495, 567, 604, and 767). Whereas positive and negative hermeneutics involved decipherment of depth— 'surface' and 'beneath the surface'—epistemological hermeneuticism flattens out understanding and perspective. There is no privileged depth to be deciphered, because everything is already decipherment. There is an ambiguous resonance between Foucault's surface flattening (everything is said) and epistemological hermeneuticism (everything is interpretation).

However, Foucault had an aversion to a whole collection of theories which included hermeneutical theories. This aversion was developed in response to his encounters with Marxist theory through his tutor Althusser and others, as well as his encounter with Hegelianism through his tutor Jean Hyppolite. His suspicion extended to psychoanalysis and hermeneutics, as evidenced in *The Order of Things* [53]. As an alternative, he developed an empirical approach to societal analysis, which he depicted as a positivism grounded in discursive archival evidence—this is proffered in *The Archeology of Knowledge* [6]. Foucault's positivism did not have or seek the rigor of the sort of positivism of the logical positivists, or of many social scientists of our time. Foucault emphasized the said over the unsaid, the visible over the invisible, and the surface over postulating anything hidden below the surface. But the way he spoke about these was sometimes more poetic than rigorous: e.g., in a 1969 interview, he says: "What I look for are not secret, hidden relations that are more silent or more profound than the consciousness of people. I try rather to define relations that are on the surface of discourse; I try to make visible what is invisible only because it is too much on the surface of things" (taken from a 1969 interview and quoted by Harcourt [17] (p. 289); Harcourt's discussion in pp. 288-89 is also quite relevant). However, as Foucault engaged in more detailed historical investigations in the 1970s and 1980s, he backed away from his earlier concern with methodology. When Hubert Dreyfus and Paul Rabinow [54] collaborated with and interviewed Foucault, it was a great opportunity for Foucault to explain his methodology, but instead he declared "The ideas which I would like to discuss here represent neither a theory nor a methodology" (p. 208). Dreyfus and Rabinow concluded that Foucault's methodology could be conceived of as an 'interpretive analytic' which, in giving a history of the present, starts with an interpretive act regarding the problematic present: "We maintain that [Foucault] is performing an interpretive act which focuses and articulates, from among the many distresses and dangers which abound in our society, those which can be seen as paradigmatic" (p. 253).

This raises a number of questions. First, one might wonder whether this sort of interpretative act of the present commits Foucault to anything methodologically substantial, rather than being something like a pre-methodological motivation—comparable to how a scientist will not bother to check whether it was science alone which led them to be motivated to explore a certain question or phenomena. Second, when Foucault examines and analyzes certain phenomena, such as in his classic studies of the clinic or of punishment, he is obviously not simply compiling quotes—what, then, is he doing? Does it not involve something other than the empirical methodology of the natural sciences and other than logical deduction—does it not involve some degree of synthesis and interpretation? How else can one make claims about societies in the past? Finally, is there a way to think of interpretation as *not* going beneath 'the surface' in any substantial way? For instance, perhaps one way to do this is to say that everything is interpretation and thus, in a sense, just as much 'surface' as anything else. However, perhaps another way to do this is to deconstruct the 'surface' metaphor or to elaborate it in other terms, such as causal terms. Foucault's opposition to a search for 'depth' could be reconciled with a hermeneutic

approach which admits that a so-called 'deep' interpretation, such as the revealing of a speaker's intention or motive or desire, is not anything causally special—something 'deep', just like an 'appearance', is likewise subject to overdetermination and power dynamics. This latter reconciliatory attempt seems akin to Slavoj Žižek's locating drive in a "materiality beyond (or, rather, beneath) the materiality located in (what we experience as) spatio-temporal reality" [55] (p. 32)—in other words, although Žižek's drive might appear to have 'depth', he claims that it is merely another sort of materiality.

I cannot do full justice to these questions here, but I raise them as starting points for different theoretical routes one could take in hermeneutically renovating Foucault and picturing that process as more of a correction, a transformation, or a reconciliation of Foucault. Additionally, in a Foucauldian spirit of theory as experimental practice, I suggest that we forge ahead with *trying out* a hermeneutically renovated Foucauldian toolbox on the CSR discursive archive, just as Foucault was more engaged with particular archives and concepts than in giving a clear account of his methodology. One doesn't need to have a fully defined and developed methodology in order to try out certain tools and lines of thought—to think and research experimentally. Arguably, I have already been trying out a hermeneutically renovated Foucauldian toolbox in Sections 3–5 above, since in those sections I have approached the corporation and CSR with Foucauldian tools but without the archival, allegedly 'positivist' grounding which often characterized Foucault's research.

One question I will address is the following: how, precisely, can we renovate Foucault's toolbox (in terms of the original elaboration of that toolbox in Section 2 and as depicted in Figure 2) so as to equip us to tackle the hermeneutic spaces of CSR? The easiest possibility for re-tooling would be to add "hermeneutic spaces" as a local *concept* for helping us understand the particular field of the corporation or of CSR. However, hermeneutist theories of different stripes suggest that hermeneutics should be understood more broadly. This could be attempted by renovating Foucault's *categories*, such as by including hermeneutics as part of the analytic grid through which we view discourse across various (or all) domains. Keeping in mind Foucault's three axes of knowledge, power, and the subject, this involves reconceiving the 'knowledge' axis. That axis would be reconceived as 'discourse' (or even 'communication'), and this axis would need to involve a hermeneutic aspect, rather than simply be knowledge oriented or truth oriented. One way to do this would be to divide the 'discourse' axis into sub-axes of interpretation, knowledge, and truth, each of which can be used as a lens for examining any given region of discourse, and different lenses will be more apt to different regions of discourse. Finally, if we take Dreyfus and Rabinow seriously, interpretation could be recognized as baked into Foucault's methodology: the meta-tool of giving a 'history of the present' involves acts of interpretation of the present as well as the past. Perhaps one could add 'problematization' as not merely a historical process, but an aspect of research: history-of-the-present research begins by taking an aspect of the present, which it then interprets as an issue of concern—as being in some way problematic and power-imbued. History-of-the-present continues by interpreting a research domain of a past period as genealogically connected to the problematic present.

Although I see reason to work with all three of the hermeneutic re-toolings which I just mentioned—the conceptual tool of hermeneutic spaces in corporation discourse, the category tool of interpretation as an aspect of discourse, and the meta tool of research-as interpretative action—my intention in the next section is to give a basic experimental start to exploring CSR hermeneutically—to argue by experimentation. Also, one does not need to incorporate all three tools to tackle the problem of the interpretative spaces of CSR. I will more directly be wielding the first and most specific tool as I turn back to the field at hand as specifically involving hermeneutic spaces, but I will revisit the second (category) tool at the end of the next section.

## 8. A Better-Equipped View of the Hermeneutic Spaces of CSR

I will start this section by adding more contour to the idea of hermeneutic spaces in CSR discourse. One way to examine hermeneutic spaces in their locality is by attending to

the conditions of their presence—the conditions under which hermeneutic gaps appear and become evident. The need to acknowledge and perform hermeneutics in a discourse may arise originally from a consciousness of its gaps and their various causes and histories, and this need intensifies when one realizes that these gaps become themselves opportunities for various tactics and uses.

The forementioned discussion of regional corporation dispositives or patterns can shed some light on the development of the hermeneutic spaces of CSR. Recall that these included surveillance, networking, human capitalization, digitization, standardization, flexibility, and consolidation. I suggest that these patterns, along with related factors such as the increasing heterogeneity of values (as facilitated in part by increasing diversity, globalization, and transnational corporations), have led to lower average levels of social fabric within companies (Porter) [45] (pp. 22–23) as well as greater hermeneutic distances. Part of what I mean by a "thinner social fabric" is that there is on average less history and context to any particular relationship or communication, so that on average parties have less shared context and background understanding of each other. A thinner social fabric combines with values-heterogeneity to produce a thinner moral fabric, meaning that there may be more differences in moral concepts and terminology, which can create gaps between meanings and standards. For example, different parties may have different conceptions of "a reliable supplier", "a responsible company", "proper prioritization", or "good crisis management". Arguably, the dark side of new media is that the cost of its cheap convenience is that it intensifies thinner social and moral fabric (in connection with its facilitation of various regional dispositives), which widens the potential for hermeneutic gaps and hermeneutic spaces.

There is an added distance which comes with the creation of organizational interlocutors and actors—the agential veiling mentioned earlier. "The company cut 5% of its workforce" has an increased distance, and increased hermeneutic space, in comparison with "The business owner cut 5% of his workers", because the organizational (corporate) actor is a nebulous collection of actors and decision-makers. There may also be further complications insofar as the actual statement issuer (such as a PR representative) may or may not be exactly saying what various executives want them to say. In the case of future-oriented statements, the power to execute or fulfill such statements may likewise rest behind a veil. When news, ratings, or certifications pertaining to a company come from a third party, then the (typically) organizational aspect of this third party likewise veils the agency and process of issuing such communication. For example, third party ratings agencies can have different criteria [38], which complexifies the interpretation of a rating.

Another feature of CSR discourse is that the stakes of the discourse have become very high—the discourse and its normalization carry power which reaches into corporate decisions, employee and potential employee decisions, consumer and client decisions, legal decisions, governmental decisions, and so forth. For such a high-powered field with heterogeneous actors, we should expect a drive for the continued development of sophisticated tactics (and resistances). Arguably, one of the main functions of the discursive field of CSR was to collect resistance to the 'incorporation', so to speak, of economic rationality. In light of all these factors, it makes perfect sense that the term 'CSR' became its own hermeneutic battleground [37,38,41,48,50].

Foucault's original conception of an "analytic grid" [56] (p. 52) referred to power and knowledge—namely, to his central 'categories'. However, I would also argue that Foucault's concepts, such as discipline, governmentality, and care-of-the-self, functioned simultaneously both as results and as grid. They did not arise until Foucault immersed himself in the field, but they then helped give him new ways to analyze and organize the field. This is how one can conceive of the first of the hermeneutic tools mentioned above—the concept of hermeneutic spaces. Once one starts to appreciate the extent of a concept like 'hermeneutic spaces', one incorporates it into one's "analytic grid". Thinking in terms of hermeneutic spaces helps us to re-view the grid of CSR.

Although this article has mainly circulated around clusters of theories and concepts, I will finally attempt to give a more earthy demonstration of what our new grid can yield. Once we view the various discourses involving CSR as hermeneutic spaces, this 'grid' or 'lens' gives us a perspective which allows us to organize elements in the field—in this case, it allows us to explore and identify hermeneutic tactics at play in hermeneutic spaces. In the following lists, I am not thinking so much of academic spaces and discourses and tactics, but mainly of everyday tactics: tactics in spaces such as corporation–investor discourse spaces, corporation–customer discourse spaces, corporation–government discourse spaces, corporation hiring discourse spaces, etc. We can divide up the hermeneutic tactics at play in these spaces as *speaker tactics* and *interpretative tactics*. These lists are a suggestive brainstorm which generalizes and greatly expands on the more specific list of company tactics in greenwashing as discussed in Freitas Netto et al. [48], e.g., for the following lists, the speaker and audience could be any party.

Some ***speaker tactics*** are mediatory, meaning they are aimed directly at bridging or diminishing the hermeneutic distance:

- Clarifying (and proactively sensing where clarification is needed);
- Cross-referencing external sources and precedents which can serve to aid interpretation;
- Transparency about evidence, uncertainty, and about various internal processes and tensions;
- Fielding questions about the meaning of an issued statement/document.

Other tactics are typically maneuvers within or utilizing hermeneutic spaces, and these may themselves be utilizable for a variety of purposes (good or bad). Some are more typically manipulative than others. These include:

- Complexification of discourse (using convoluted language or jargon);
- Overloading discourse (e.g., creating a 120-page contract which no one will read carefully);
- Specific and contextual selection (giving hand-picked examples or data while avoiding or hiding other specifics, contexts, or broader trends);
- Vague generalities (while avoiding or hiding specifics);
- Redirection ("Look, I don't have the information you request in front of me, let me try to get back to you later, or you can call our other department. But what I can say is that we are very responsible about these other matters!");
- Statistical presentation;
- Ethos presentation ("We here at X foster a culture of excellence and transparency");
- Story telling/narration ("Five years ago we were unaware of the inhumane conditions of our supply chain, but when we heard about this recently, we started taking steps to fix the situation").

*Interpretative (or receiving) tactics* include the following positive tactics:

- Charity of interpretation (interpreting that the speaker did not mean the offensive or thoughtless claim which they might have meant);
- Benefit of the doubt (interpreting that the speaker is speaking honestly or transparently about what cannot be verified independently);
- Trust (interpreting that the speaker is not hiding ulterior motives and intentions which could harm various parties).

On the other hand, suspicious interpretative tactics include:

- Suspicion (assuming, until independently investigated, that the speaker is hiding some information, motives, or intentions which are relevant to the discourse);
- Discounting (suspecting that the reported statements are not as relevant or important as the speaker is trying to suggest);
- Unmasking (speculating as to the underlying motives behind the speaker's statements).

Investigatory interpretative tactics include:

- Direct investigation (directly asking the speaker for clarification, more information, motives, or intentions);

- Comparing sources and precedents (e.g., checking what the speaker said in another context and whether that was verified or confirmed);
- Independent research about what is stated or the source of the statement.

Propagation tactics include:

- Quoted transmission ("Party A reported X");
- Selective interpretation (or selective transmission, or selective "memory");
- Taking statements out of context or repackaging their context;
- Stretching words or concepts to encompass situations and objects outside the original speaker's purview.

Even if these lists of tactics are neither complete nor properly investigated, I claim that they are clearly not the same sorts of tactics which Foucault was uncovering as he investigated knowledge-oriented or truth-oriented discourse. I read this as confirmation of the need to reconceive of the axis or category sometimes labeled as "knowledge" and sometimes as "discourse" (the second version of re-tooling suggested in the last section). Having surveyed the above tactics, I want to suggest the following idea: we can conceive of discourse as the proper axis, and then recognize that different regions of discourse are oriented differently and involve different tactics. Knowledge-oriented regions of discourse and truth-oriented regions of discourse will have different tactics from hermeneutic spaces of discourse. Power circulates differently—by way of different tactics—in these differently oriented regions of discourse.

However, some regions of discourse may not fall so simply into one of the three orientations. The discussion of the definition of 'CSR' itself is one such region. It is a mixed region. In the classroom, it appears more like an item of knowledge to be tackled with knowledge-tactics such as reasoning, argumentation, debate, normalization, normalization critique, and specialization. However, a corporate website or advertisement functions more as a hermeneutic site.

## 9. Conclusions

I started out this paper with a rework of the way Koopman and Matza suggest Foucault be put to work. One of the key suggestions I made was to see Foucault's own approach, and the ways we can utilize Foucauldian elements, in a more flexible way—in particular, to be broadly open to reworking the Foucauldian toolbox. This provided the backdrop for allowing the field of the corporation and the field of CSR to speak both to the helpfulness and insufficiency of Foucault's traditional tools. I claimed that the dispositive is a helpful tool, but it could be more useful if it were stretched beyond its original usage. I also claimed that the hermeneutic spaces of CSR suggest the insufficiency of Foucault's discursive tools, and that they call for a renovation of Foucauldian tools so as to include hermeneutic tools. I then suggested three possible ways to retool—at the conceptual level, the category level, and the meta-tool level. Finally, I demonstrated the utility of hermeneutic re-tooling in uncovering tactics at play in the hermeneutic spaces of CSR.

The idea of 'CSR' as a mixed region of discourse—partly knowledge oriented and partly hermeneutic and involving both knowledge tactics and hermeneutic tactics—raises the question of whether other regions of discourse are similarly 'mixed'. For instance, are most or all political regions of discourse similarly mixed, involving a similar complicated confusion of tactics? When one changes a tool, one may change one's way of seeing the world more than one originally realized.

**Funding:** This research received no external funding.

**Institutional Review Board Statement:** Not applicable.

**Informed Consent Statement:** Not applicable.

**Data Availability Statement:** Not applicable.

**Acknowledgments:** Thanks especially to the editors for extensive, helpful feedback in the development of this article.

**Conflicts of Interest:** The author declares no conflict of interest.

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
