# Peer review of "CSR and the Hermeneutical Renovation of Foucault’s Toolbox"

_sustainability, doi:10.3390/su15054682_

Round 1

Reviewer 1 Report

The intellectual challenge and joy(?) experienced by contrasting the perspectives in one field of research (Foucault epistemology) with another field of research considered more superficial - and therefore referred to as a "subfield" of this research (CSR), demands an anchoring in a well documented and representative account of the sources. In this paper the validity and reliability of the "Foucauldian approach to the corporation" rests on the paper by Koopman & Matza's. The presentation of CSR is sketchy and serves as input to our understanding of Foucault. The version of Foucault in order to analyse the CSR discourse, appears as an outcome of a critique of Koopman & Matza. This is original and seems to be a useful approach considering the many presentations of Foucault available in this setting.  The critique itself (resisting universality, the inflexible one-way directionality, and the lack of the activists' role) seems well argued. 

CSR is seen as a space of normative discourse about corporations in which "various dispositives verbally contend and are contended with". And as a subfield of discourse, CSR is distinguished by being related to a "plurality of historical strands", not to any singular progression associated with the appearance of the modern corporation. 

The "hermeneutic space" surrounding CSR practices is well described and this leads us to the list of "speaker tactics" and "interpretive tactics" presented listed on pages 15-16. These are attributes of power and vary between regions and, according to the author, reveals the insufficiency of Foucaults' discursive tools.

- - -

In my mind this paper casts more light on Foucault, less on CSR. This is interesting and worth publishing as long as this kind of philosophical approaches are acceptable in a journal covering the sustainability field.

Author Response

Most of the review is positive, which I really appreciate.

"The presentation of CSR is sketchy"

This is basically the only criticism in this review. In response, I added a variety of sources suggested directly by the editors and integrated them in various sections. These include: 

Thompson, P., & Ackroyd, S. (1995) All quiet on the workplace front? A critique of recent trends in British industrial sociology. Sociology, 29(4): 615–633 

Wagler, R. (2009) Foucault, the Consumer Culture and Environmental Degradation. Ethics Place and Environment, 12(3): 331-336.  

de Freitas Netto, S.V.; Sobral, M.F.F.; Ribeiro, A.R.B. et al. (2020) Concepts and Forms of Greenwashing: A systematic review. Environmental Sciences Europe, 32(19). 

Abela, M. (2022) A New Direction? The “Mainstreaming” of Sustainability Reporting. Sustainability Accounting, Management and Policy Journal, 13(6): 1261-1283. 

Silva-Casteñada, L. and Trussart, N. (2016) Sustainability Standards and Certification: Looking Through the Lens of Foucault's Dispositif. Global Networks, 16(4): 490–510. 

Elena Escrig-Olmedo et al. (2019) Rating the Raters: Evaluating how ESG Rating: Agencies Integrate Sustainability Principles. Sustainability, 11: 915-932. 

Zizek, S. ‘Seven Veils of Phantasy’:  https://www.taylorfrancis.com/chapters/edit/10.4324/9780429476402-8/seven-veils-fantasy-slavoj-%C5%BEi%C5%BEek 

Reviewer 2 Report

Thank you for giving me the opportunity to review the article „CSR and the Hermeneutical Renovation of Foucault’s Toolbox”. Below my remarks:

- Author has chosen an interesting and current topic on Renovation of Foucault’s Toolbox;

- Abstract lacks a clearly defined purpose for the article;

- I suggest separate the “Introduction: Foucault’s Toolbox” section into two sections: Introduction and Literature review;

- In the final part of the Introduction, please describe synthetically what the individual sections of the article contain;

- Literature review is correct, while it could still be expanded;

- In conclusion, Author should also describe the limitations of the study and the directions of future research.

- References should be properly prepared. E.g. Kalmykova, Y.; Harder, R.; Borgestedt, H.; Svanäng, I. Pathways and Management of Phosphorus in Urban Areas. J. Ind. Ecol. 2012, 16, 928–939.

Author Response

- Abstract lacks a clearly defined purpose for the article;

The abstract--and especially the introduction--were revised to help address this.

- I suggest separate the “Introduction: Foucault’s Toolbox” section into two sections: Introduction and Literature review;

I did this, except I kept my own title for the second section.

- In the final part of the Introduction, please describe synthetically what the individual sections of the article contain;

I did this.

- Literature review is correct, while it could still be expanded;

I added sources suggested by the editors.

- In conclusion, Author should also describe the limitations of the study and the directions of future research.

I attempted to acknowledge limitations in the revised introduction (section 1). There is some revision in 6 that is also relevant, e.g., where I say "while I don’t pretend to make much headway into such problems of subjectivization."

Reviewer 3 Report

This is an interesting study and the authors have performed an in-depth review of the past literature. However, in my opinion, the manuscript has some shortcomings.

The abstract section need revision, future tense was used. Because the author already completed the study. Please avoid the use of I, me in writing.

The introduction section has lack of study motivation. Some of the references are missing and incomplete. References are too old. please use the latest references from 2018-2020. The flow of study from one to section to another section are missing. A through proofreading and editing are required.

Author Response

The abstract section need revision, future tense was used. Because the author already completed the study. Please avoid the use of I, me in writing.

These are not standards I am used to following in my field (philosophy), and the editors did not seemed concerned about them. However, I did modify the abstract itself in line with these standards (present tense, no use of "I").

The introduction section has lack of study motivation. Some of the references are missing and incomplete. References are too old. please use the latest references from 2018-2020. The flow of study from one to section to another section are missing. A through proofreading and editing are required.

The introduction is revised. There are a bit more connecting paragraphs to address flow. The editors helped my to include a richer array of references, which now include the following: 

Thompson, P., & Ackroyd, S. (1995) All quiet on the workplace front? A critique of recent trends in British industrial sociology. Sociology, 29(4): 615–633 

Wagler, R. (2009) Foucault, the Consumer Culture and Environmental Degradation. Ethics Place and Environment, 12(3): 331-336.  

de Freitas Netto, S.V.; Sobral, M.F.F.; Ribeiro, A.R.B. et al. (2020) Concepts and Forms of Greenwashing: A systematic review. Environmental Sciences Europe, 32(19). 

Abela, M. (2022) A New Direction? The “Mainstreaming” of Sustainability Reporting. Sustainability Accounting, Management and Policy Journal, 13(6): 1261-1283. 

Silva-Casteñada, L. and Trussart, N. (2016) Sustainability Standards and Certification: Looking Through the Lens of Foucault's Dispositif. Global Networks, 16(4): 490–510. 

Elena Escrig-Olmedo et al. (2019) Rating the Raters: Evaluating how ESG Rating: Agencies Integrate Sustainability Principles. Sustainability, 11: 915-932. 

Zizek, S. ‘Seven Veils of Phantasy’:  https://www.taylorfrancis.com/chapters/edit/10.4324/9780429476402-8/seven-veils-fantasy-slavoj-%C5%BEi%C5%BEek 

Reviewer 4 Report

My Comments

This work investigates “CSR and the Hermeneutical Renovation of Foucault’s Toolbox” I am reasonably familiar with the literature on the topic. I applaud your efforts to work on the topic but unfortunately, in my opinion, the discussion given in this paper is shallow, and I have a hard time seeing which specific conclusions could be drawn from it. Please see below my concerns in detail.

1.     First the author(s) must know what should be included in the abstract. The abstract is unnecessarily lengthy by adding some irrelevant information.

2.     I believe the introduction is quite comprehensive. Despite this, it fails to state the potential research gaps and research objectives and outline the main research questions. In the paper, there is lack of consistency and continuity. I suggest the authors that they must introduce the topic in the introduction section by incorporating the gap in the second last paragraph by adding the following information). In the introduction section in the second last paragraph as well as in the abstract section, the following information must be included. , such as What the author(s) want to know? (Purpose), Why do they want to know? (gap), How do they want to know? (Methodology), What are the outcomes? (Results),What are the developments of the study? (Contribution), What are the limitations of the study? (future research direction).

3.     In my opinion the literature review section is quite comprehensive but I did not see any theoretical background. Although, this is one of the main limitations of this study. For that, you can refer to the studies such as https://doi.org/10.3390/su14116917.

4.     Analysis section is very hard to follow.

5.     Discussion should be included compare and contrast with previous studies, discussion of the results, theoretical contribution, practical implication, and future research direction Please refer to this study https://doi.org/10.3390/su142416336.

6.     Finally, if the authors are willing to amend the paper, I will be happy to review it again.

Author Response

1. First the author(s) must know what should be included in the abstract. The abstract is unnecessarily lengthy by adding some irrelevant information.

The abstract was revised.

  1. I believe the introduction is quite comprehensive. Despite this, it fails to state the potential research gaps and research objectives and outline the main research questions. In the paper, there is lack of consistency and continuity. I suggest the authors that they must introduce the topic in the introduction section by incorporating the gap in the second last paragraph by adding the following information). In the introduction section in the second last paragraph as well as in the abstract section, the following information must be included. , such as What the author(s) want to know? (Purpose), Why do they want to know? (gap), How do they want to know? (Methodology), What are the outcomes? (Results),What are the developments of the study? (Contribution), What are the limitations of the study? (future research direction).

To some extent, I tried to clarify the paper's structure better by adding a new introduction section. However, I believe this reviewer's standards for structuring a paper arise from a field which is different than my own training (philosophy). Since my aim was to write a philosophy paper rather than a proper social science paper, I did not exactly follow the structure advised here.

  1. In my opinion the literature review section is quite comprehensive but I did not see any theoretical background. Although, this is one of the main limitations of this study. For that, you can refer to the studies such as https://doi.org/10.3390/su14116917.

I don't understand the sort of theoretical background that is requested, however, the editors did suggest a variety of new sources to add, and which I did add. However, those sources were mostly pertaining to new research in CSR and related discourse. The new sources include

Thompson, P., & Ackroyd, S. (1995) All quiet on the workplace front? A critique of recent trends in British industrial sociology. Sociology, 29(4): 615–633 

Wagler, R. (2009) Foucault, the Consumer Culture and Environmental Degradation. Ethics Place and Environment, 12(3): 331-336.  

de Freitas Netto, S.V.; Sobral, M.F.F.; Ribeiro, A.R.B. et al. (2020) Concepts and Forms of Greenwashing: A systematic review. Environmental Sciences Europe, 32(19). 

Abela, M. (2022) A New Direction? The “Mainstreaming” of Sustainability Reporting. Sustainability Accounting, Management and Policy Journal, 13(6): 1261-1283. 

Silva-Casteñada, L. and Trussart, N. (2016) Sustainability Standards and Certification: Looking Through the Lens of Foucault's Dispositif. Global Networks, 16(4): 490–510. 

Elena Escrig-Olmedo et al. (2019) Rating the Raters: Evaluating how ESG Rating: Agencies Integrate Sustainability Principles. Sustainability, 11: 915-932. 

Zizek, S. ‘Seven Veils of Phantasy’:  https://www.taylorfrancis.com/chapters/edit/10.4324/9780429476402-8/seven-veils-fantasy-slavoj-%C5%BEi%C5%BEek 

  1. Analysis section is very hard to follow.

I'm not sure which section this refers to, but anyways, I would agree that my writing is not the most perspicuous, and I tried to follow a variety of specific editorial suggestions to mitigate this issue.

  1. Discussion should be included compare and contrast with previous studies, discussion of the results, theoretical contribution, practical implication, and future research direction

As mentioned above, I attempted to integrate a variety of references to various recent research in CSR. 

Round 2

Reviewer 4 Report

I am sorry to say I am completely unsatisfied from the revised version of the manuscript. The author did not try to incorporate any single of my comment. always the author mentioned I am not sure. What is this????? 

Author Response

The author did not try to incorporate any single of my comment.

I suggested that our standards for journal writing are different, which is why I mostly didn't follow this reviewer's suggestions. I wrote a philosophy essay, not a social science article.

Round 3

Author Response

Dear Authors, If it a philosophy essay, not a social science article then why you mentioned This article aims to examine Michel Foucault’s conceptual toolbox?

Philosophy, according to its name, seeks wisdom. The "social science article" is not the main way philosophers have been writing and seeking wisdom (for millennia). This article intends to be a sort of philosophy essay. Also, mdpi has a different description of an "essay" from that of an "article"--see https://www.mdpi.com/about/article_types

The paper should be good and compressive for that you need to search different articles. As you have searched and reviewed only 56 articles and books etc. Therefore, by adding the following relevant literature from different authors can significantly improve the quality of the manuscript. It will increase the no of references and will add value to your article. I hope you will consider these articles.

These articles seem to be very detailed empirical work, but they don't appear to be about Foucault or CSR, so I don't see their relevance to this paper.